# New Biostimulants Screening Method for Crop Seedlings under Water Deficit Stress

David Jiménez-Arias [1,2,*], Sarai Morales-Sierra [3], Andrés A. Borges [2], Antonio J. Herrera [4] and Juan C. Luis [3,*]

1. Investigador Principal Convidado, ISOPlexis, Centro de Agricultura Sustentável e Tecnologia Alimentar, Campus Universitário da Penteada, 9020-105 Funchal, Madeira, Portugal
2. Chemical Plant Defence Activators Group, Department of Agrobiology, IPNA-CSIC, Avenida Astrofísico Francisco Sánchez 3, P.O. Box 195, 38206 La Laguna, Tenerife, Canary Islands, Spain; aborges@ipna.csic.es
3. Grupo de Biología Vegetal Aplicada (GBVA), Departamento de Botánica, Ecología y Fisiología Vegetal-Facultad de Farmacia, Universidad de La Laguna, Avenida. Astrofísico Francisco Sánchez s/n, 38071 La Laguna, Tenerife, Canary Islands, Spain; smorales@ull.edu.es
4. Natural Product Synthesis Group, Department of Chemistry of Bioactive Natural and Synthetic Products, IPNA-CSIC, Avenida. Astrofísico Francisco Sánchez 3, P.O. Box 195, 38206 La Laguna, Tenerife, Canary Islands, Spain; ajherrera@ipna.csic.es
* Correspondence: david.j.a1983@gmail.com (D.J.-A.); jcluis@ull.edu.es (J.C.L.)

**Abstract:** Biostimulants can be used in many crops growing under water deficit conditions at the seedling stage. This study used tomato, Solanum lycopersicum L., seedlings growing in commercial 150-cell trays as an experimental setup to reproduce mild drought stress effects. The method showed significant reductions in seedling growth and RGR (25%) after a seven-day experiment. Gas exchange parameters (Pn, Gs and E) had significantly lower values (30–50%) than the control seedlings. Stress-related metabolite, ABA, exhibited a significant accumulation in the tomato seedlings (24 h), consistent with SINCED2 gene expression. Proline levels were twice as high in the water-deficit treated seedlings, remaining at this level until the end of the experiment. However, total carbohydrates were significantly lower in water-deficit treated seedlings. Qualitative and quantitative analysis suggested that using the variable 'seedling biomass accumulation' could simplify the methodology. Twelve different biostimulants were assayed, implementing this simplification, and all of them showed higher biomass accumulation in the treated seedlings than in the non-treated ones under water deficit. Among them, putrescine, spermine and spermidine were the most effective. The method is adjustable to different biostimulant volumes (1, 3 and 5 mL; 1 mM BABA), with no significant differences between the treatments.

**Keywords:** biostimulant; evaluation; abiotic stress; growth promoters; model system; drought

## 1. Introduction

Crop productivity is highly dependent on irrigation management and water quality. Consequently, global climate change is a threat to crop production. In the words of M. Mizutori—UN special representative for disaster risk reduction—"Drought is a hidden global crisis, at risk of becoming the next pandemic if countries do not take action on water and land management and at the same time tackle the climate emergency" [1]. Indeed, drought losses reached USD 124 billion during the 1998–2017 period and were suffered by more than 1.5 billion people [2]. With the increasing global population, estimated to reach up to 9.8 billion by 2050, the proportion of undernourished people will increase every year. It is expected to be more than 3 billion by the end of this century [3]. Therefore, a substantial increase in food production is needed, raising more than one concern about the use and availability of water [4]. Future climate predictions point out water shortage as one of the leading global concerns in coming years, severely affecting agricultural systems, crop yield and product quality [5]. Irrigation water needs will increase by more than 50%

in developing regions and 16% in developed ones; consequently, creating intense pressure on freshwater resources [6].

Plant responses to water deficit depend on several factors, including the duration and extent of the stress and the possibility of activating specific tolerance mechanisms [7]. Long-term drought causes physiological and metabolic changes that include cell turgor loss, water imbalances and decline in gas-exchange parameters [8]. Plants cope with drought by using various perception signals, and among them, abscisic acid (ABA) is common [9,10]. Biosynthesis of ABA induces stomatal closure [11] and proline accumulation through the ABA-dependent pathway [12]. In this regard, expression of the 9-cis-epoxycarotenoid dioxygenase (NCED) gene is critical in ABA biosynthesis, *SlNCED2* being important in ABA accumulation under stress [13]. Future crop productivity must find new strategies, and new biostimulants are a valuable option. Biostimulants are nowadays a hot topic [14], with a high potential for improvement in decades to come, especially in crop productivity under abiotic stress [14–22]. Indeed, they can improve yield in the field under water deficit stress [23–25], and are becoming an exciting tool to cope with environmental stresses in world climate change scenarios [26]. Biostimulants also have substantial economic implications; in fact, it is estimated that the global market of these products will reach USD 4.2 billion by 2025 [27].

The current definition of plant biostimulants in the EU regulation (2019) is: "A product that stimulates plant nutrition processes independently of the product's nutrient content, with the sole aim of improving one or more of the following characteristics of the plant or the plant rhizosphere: (a) nutrient use efficiency; (b) tolerance to abiotic stress; (c) quality traits; or (d) availability of confined nutrients in the soil or rhizosphere" [28]. Raw materials for biostimulants come from different sources, thus allowing the end product to be a solid [25], liquid [29] or even a gas formulation [30]. This heterogeneity in sources and end products has led to varying biostimulant classifications over the years. However, Du Jardin established a well-known and accepted classification that grouped biostimulants into six different classes [31]: (i) seaweed and botanical extracts; (ii) chitosan and other biopolymers; (iii) beneficial fungi and bacteria; (iv) protein hydrolysates and N-containing compounds; (v) inorganic compounds; (vi) humic and fulvic acids.

Biostimulant development and selection should use plants' physiological, metabolic and genetic mechanisms to cope with drought. In the bibliography, many studies positively correlate these mechanisms against drought tolerance: higher gas exchange parameters to increase plant adaptation under water deficit [32–34], proline [35–37] and ABA accumulation [38–41]. However, beyond specific biochemical or physiological traits, new biostimulant design formulations require a suitable and straightforward testing protocol to describe their morpho-physiological effects on plants, under the imposed biotic or abiotic stress [42]. In this regard, a range of methods and model organisms have been used to study biostimulants' effects under stress conditions, such as: (i) in vitro yeast growth followed by a seed germination stage [43], (ii) Arabidopsis germination and rosette growth treated via a High-Throughput Screening platform [44], (iii) High-Throughput Plant Phenotyping linked with Metabolomics [42], and (iv) using inducible reporter lines such as ABA-inducible luciferase to search for agonist molecules [45]. These procedures are new options to test and study biostimulants; however, they need a high-tech upgrade not accessible to all laboratories or companies.

This study aimed to develop a low-cost, straightforward and accessible procedure to test biostimulant compounds and formulations on a specific crop, under controlled environmental growing conditions. The described method reproduces the same physiological, biochemical and molecular responses of fully developed plants subjected to water deficit, but uses crop seedlings instead. In addition, the method simplifies the technology and variables needed in biostimulant selection treatments and reduces the assessment time to seven days. This methodology can be used before any field study on many different crop seedlings, and summarises our own experience in new biostimulant development [46,47].

## 2. Materials and Methods

### 2.1. Plant Material and Experimental Conditions

Standard 150-cell tomato seedling trays were bought at a local commercial crop nursery. *Solanum lycopersicum* L. seeds were sown in the trays, using an automatic sowing machine to ensure germination and growth uniformity, until the two true-leaf stage (two weeks). Only size-uniform, well-rooted and disease-free seedlings were used in the experiments (Figure 1 explains the treatment distribution and the cell tray characteristics). Seedling trays were transferred to a growth chamber with controlled conditions: temperature 20–27 °C, photoperiod 16–8 h, humidity 60–75%, and irradiance 300 µmols m$^{-2}$ s$^{-1}$. All the plants were watered with a half-strength Hoagland [48].

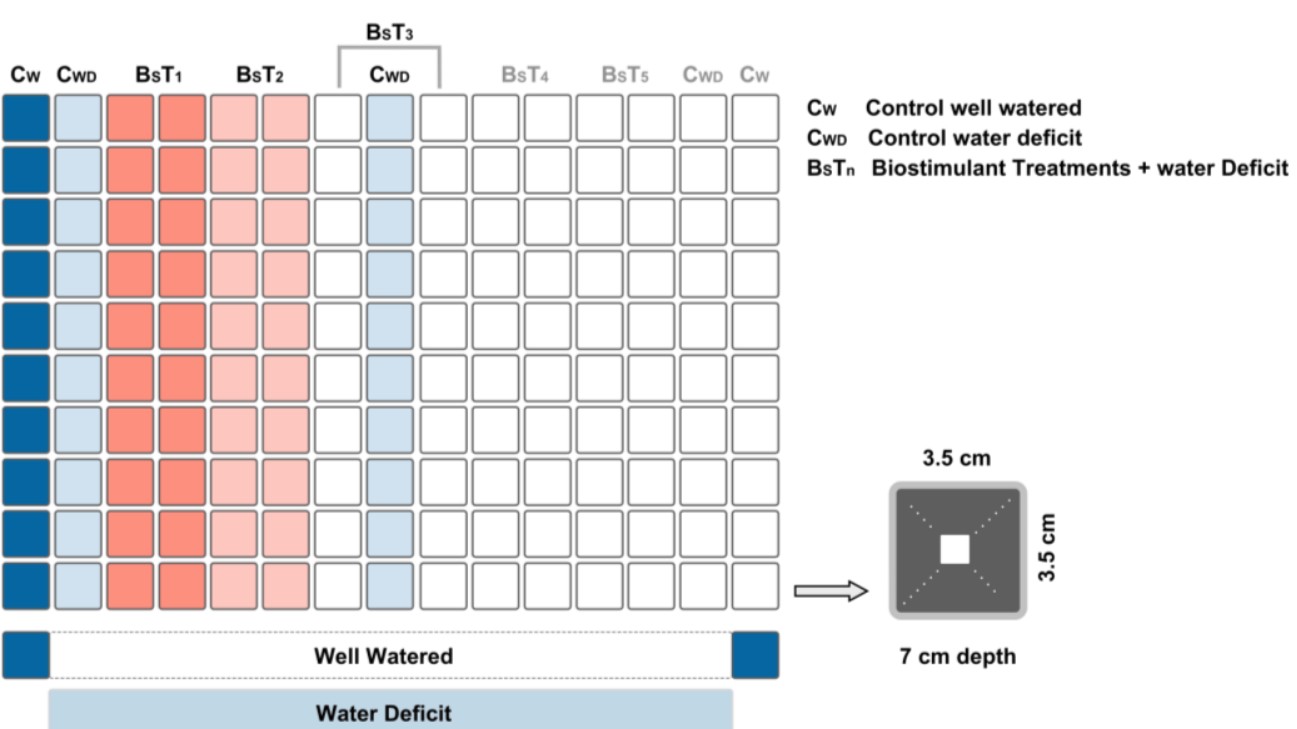

**Figure 1.** Experimental treatments distribution (20 seedlings per treatment) and cells dimensions of a 150-cell seedling tray.

Two different experiments were conducted using the 150-cell trays (Figure 2). The first assessed if the known water deficit effects on fully developed plants were reproduced in water-deficit treated tomato seedlings. Every day, the field water capacity per seedling cell-tray was calculated. This value was used to water control seedlings, and water-deficit seedlings received 50% of this volume. The water-deficit experiments ended on the seventh day, and growth, biochemical and genetic data were collected as shown in Figure 2. The second experiment evaluated different biostimulant treatments of the tomato seedlings arranged in the seedling trays as shown in Figure 1, measuring only seedling growth with a minimum of 20 plants per treatment, again over seven days. As in the previous case, field water capacity guided the watering of the controls (100%) and the treated seedlings (50%). All the experiments were performed in triplicate.

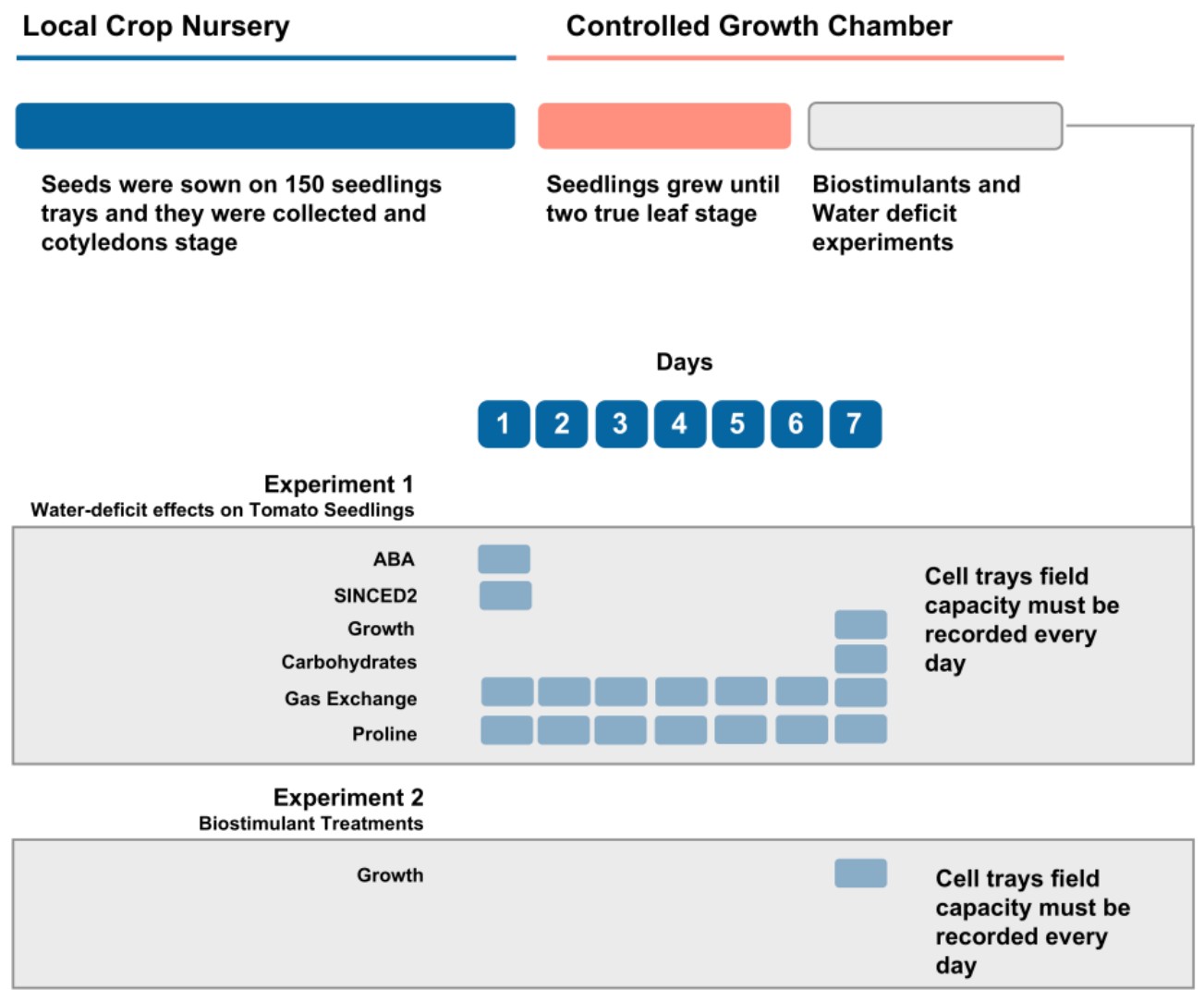

**Figure 2.** Timeline from local crop nursery to laboratory growth chambers and variables data collection in experiments with biostimulants under water deficit. Experiment 1 assessed wa-ter-deficit, reproducing its known effects on tomato seedlings. Experiment 2 validated the sim-plified method, assessing only the growth in biostimulant-treated tomato seedlings subjected to water-deficit.

*2.2. Biostimulants Used in the Study*

Biostimulants were selected according to their known effects on plant tolerance against water deficit. Pure molecules were preferred to extracts, although a complex mixture was also used—a seaweed (macroalgal) extract known as Kelpak® [15]. Pure inorganic molecules such as sodium and potassium silicate [49] were also assayed in the study. All the chemicals used in the experiment were purchased from Aldrich Chemical Co. (St. Louis, MO, USA). Organic N-containing molecules such as polyamines [50] (putrescine, spermi-dine, norspermidine, spermine and norspermine), proline [51], pyro-glutamic acid [25], betaines [23] (glycine-betaine) and melatonin [52] were also evaluated. Pure inorganic and organic molecules were solubilised in half-strength Hoagland solution and pH adjusted to 6, except for the seaweed extract, which was applied according to the manufacture's recom-mendations (Table 1). The control treatments received the same volume of half-strength Hoagland solution.

**Table 1.** Biostimulants used in the experiments and their concentrations.

| Group | Molecule Name | Cas Number | Abbreviation | mM/(%) * |
|---|---|---|---|---|
| Hormones | Putrescine | 110-60-1 | Put | 1 |
| | Espermidine | 124-20-9 | Esp | 0.1 |
| | Norspermidine | 56-18-8 | N-Esp | 1 |
| | Spermine | 71-44-3 | Spm | 1 |
| | Norspermine | 4605-14-5 | N-Spm | 0.1 |
| | Melatonin | 73-31-4 | Mel | 1 |
| Amino Acids | Proline | 147-85-3 | Pr | 1 |
| | Pyroglutamic acid | 98-79-3 | Pyg | 1 |
| | B-Amino Butiric Acid | 541-48-0 | BABA | 1 |
| Betaines | Glycine-Betaine | 590-46-5 | GB | 0.1 |
| Inorganic | Sodium siliciate | 6834-92-0 | Na-Si | 1 |
| | Potassium silicilitate | 1312-76-1 | K-Si | 2 |
| Seawed extract | Kelpak® (BASF; Ludwigshafen; Germany) | | Kelp | (5) * |

* means %.

### 2.3. Growth and Gas Exchange Measurements

Seedling biomass analyses were conducted at the beginning and end of the water-deficit period (7 days). Seedlings were extracted from the cell trays, and the roots were carefully washed under water to remove peat. Seedlings were oven-dried at 70 °C for three days. The Relative Growth Rate (RGR) was estimated as described in [53], following the formula: RGR = $(\ln W_2 - \ln W_1)/(t_2 - t_1)$, where $W_1$ and $W_2$ are seedling dry weights at times $t_1$ and $t_2$ (the beginning and end of water deficit, respectively). The leaf Relative Water Content (RWC) was calculated as follows: RWC = (FM − DM)/(TM − DM) (FM—fresh mass; DM—dry mass; TM—turgid mass). Seedling water-use efficiency was determined as described by Medrano et al. [54].

During the experiments, gas exchange analyses were carried out on the fully expanded leaves (N = 20), as described elsewhere [55]. Photosynthesis (Pn), stomatal conductance (Gs) and the transpiration rate (E) were measured on the attached leaves with a portable Infrared Gas Analyser (LCPro, BioScientific Ltd., Hoddesdon, UK). Water use efficiency (WUE) values are the ratios between Pn and E. The measurements were taken at ambient $CO_2$ concentration, a photosynthetic photon flux density (PPFD) of 1000 µmol m$^{-2}$ s$^{-1}$ (optimised with a light curve), and a cuvette air flow rate of 500 mL min$^{-1}$.

### 2.4. Biochemical and Molecular Measurements

Proline leaf accumulation was analysed daily, and total carbohydrates at the end of the experiment (N = 6). The concentrations were determined following the procedures described in reference [55]. The Abscisic Acid (ABA) biosynthesis gene, *SlNCED2*, expression was monitored in the leaf samples from each water regime (well-watered and water-deficit plants) 10, 24 and 72 h after the imposed water deficit. Real-time PCR analysis was performed for *SlNCED2* quantification, as described by [56], using specific primers [13] (primers sequences in Table S1). EF-1$\alpha$ was used as a housekeeping gene. ABA accumulation on the seedlings' leaves was analysed at 6, 12 and 24 h after the imposed water deficit. HPLC analysis and quantification was performed as described in reference [57]. Two pools of ten plants per condition were used, and triplicate HPLC analyses and quantification from each pool were performed.

### 2.5. Statistical Procedure

One-way ANOVA tests (Duncan's post hoc) were applied to analyse the significance of the differences between the experimental groups. Correlation analyses were used to understand the relationship between the variable biomass and the other variables. All statistical studies were performed on IBM-SPSS24 software.

## 3. Results

### 3.1. Seedlings Growth and Gas Exchange Measurements under Water-Deficit

Seedling dry weight and RGR showed, on average, a 25% reduction at the end of the seven days of the experiment for both variables. Water-deficit treated seedlings also exhibited significant differences in their RWC and WUE, compared to the well-watered plants, reaching 16% and 50% reductions, respectively (Figure 3). Gas exchange measurements followed a similar trend. Net Photosynthesis (Pn), Stomatal Conductance (Gs), and the Transpiration Rate (E) displayed 43%, 50% and 30% reductions in their respective values (Figure 4). These differences from well-watered control plants were statistically significant from the second day of the experiment onwards. In addition, intrinsic water use efficiency (WUEi) was higher in the well-watered plants during all the experiments. In contrast, the opposite behaviour was seen in instantaneous water use efficiency (WUEinst), with higher levels in water-deficit treated seedlings until the seventh day, when they became equal (Figure 5).

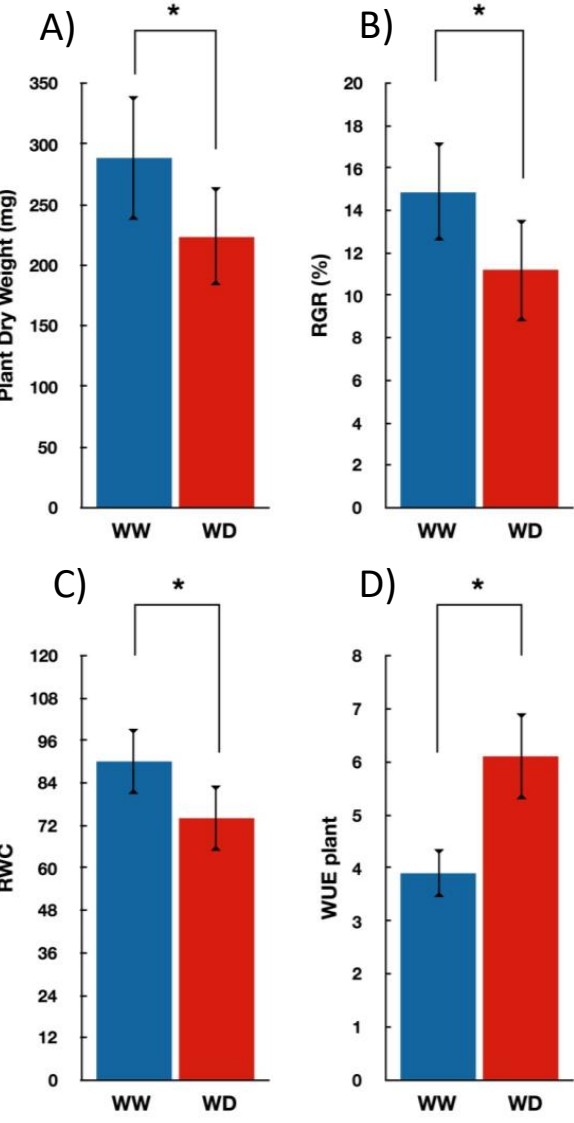

**Figure 3.** (**A**) Seedlings DW (dry weight), (**B**) RGR, (**C**) RWC and (**D**) WUE$_{plant}$ on the seventh day of the experiment. Blue bars represent well-watered seedlings; red bars represent water-deficit treated seedlings. * means *p* value < 0.05.



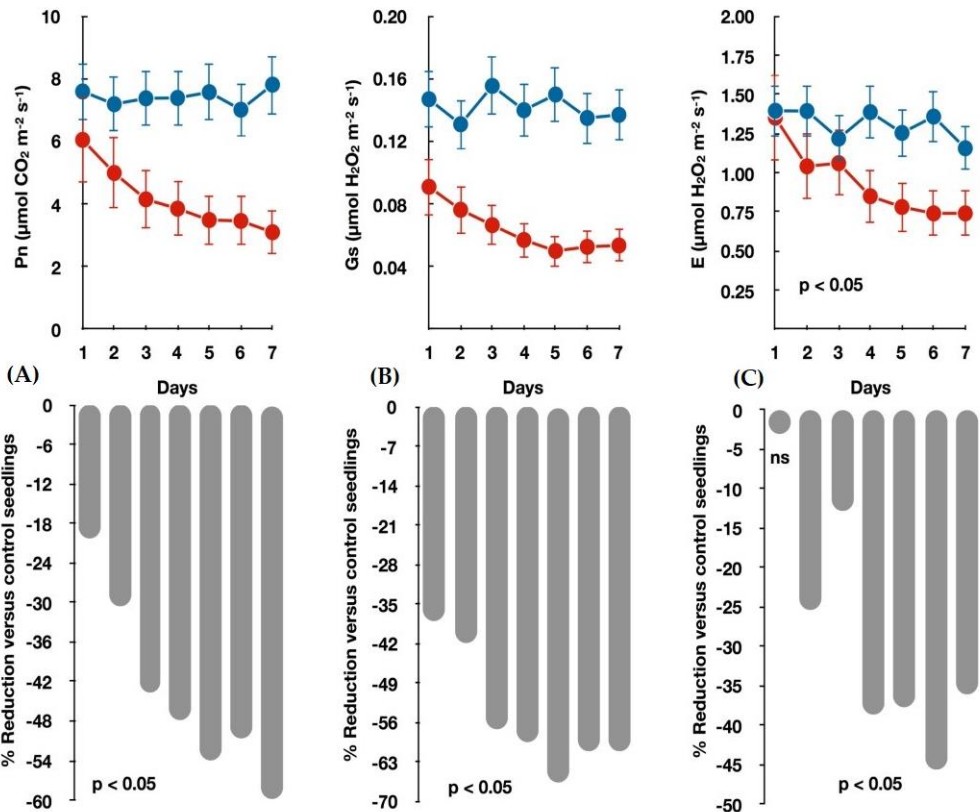

**Figure 4.** (**A**) Daily seedlings Photosynthesis (Pn), (**B**) Stomatal Conductance (Gs) and (**C**) Transpiration rate measures and their respective % drop in comparison with control seedlings. Blue lines represent well-watered seedlings; red lines represent water-deficit treated seedlings. Statistical differences, $p$ value $< 0.05$, between treatments were observed from the second day of the experiment. ns, not significant.

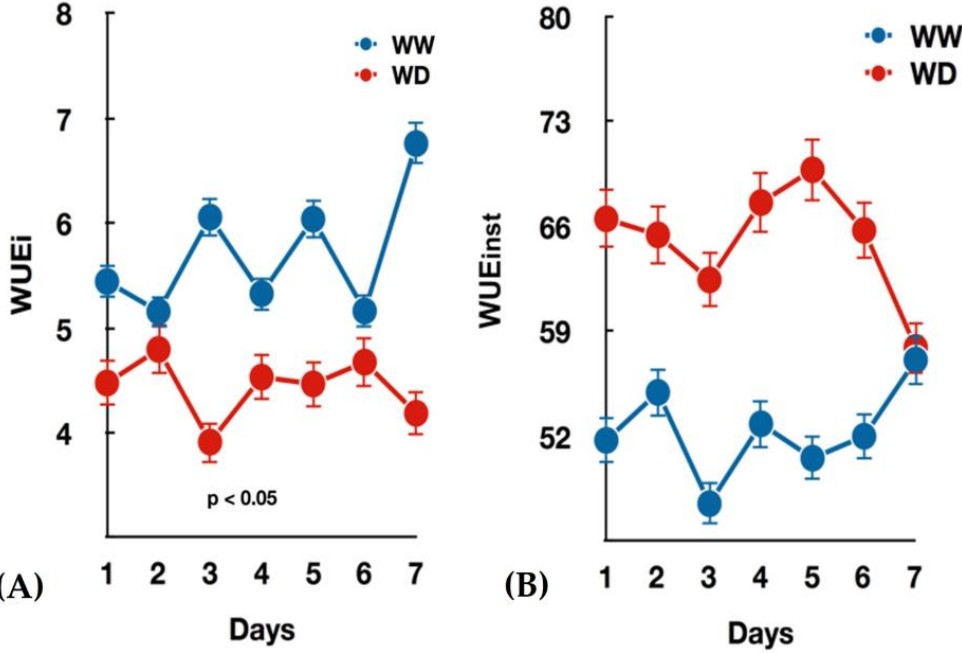

**Figure 5.** Daily (**A**) seedlings intrinsic water use efficiency (WUE$_i$) and (**B**) instantaneous water use efficiency (WUE$_{inst}$). Blue lines represent well-watered seedlings; red lines represent water-deficit treated seedlings.

### 3.2. Seedlings Biochemical and Molecular Changes under Water Deficit

The seedlings subjected to water-deficit showed higher proline levels than control plants during the first two days of the experiment, despite not being statistically significant. Only after three days were proline values statistically significant in the water-deficit treatments, remaining twice as high as the well-watered seedlings until the seventh day of the experiment (Figure 6A,B). A similar trend continued in the ABA leaf accumulation and *SlNCED2* gene expression during the 24 h period. In fact, 10 h after water-deficit started, a significant increase in *SlNCED2* expression was detected, which remained upregulated during the next 24 h. However, two days later at 48h, the gene expression was downregulated. These gene expression patterns were translated into a high ABA accumulation after 24 h in the water-deficit treated seedlings (Figure 6C,D).

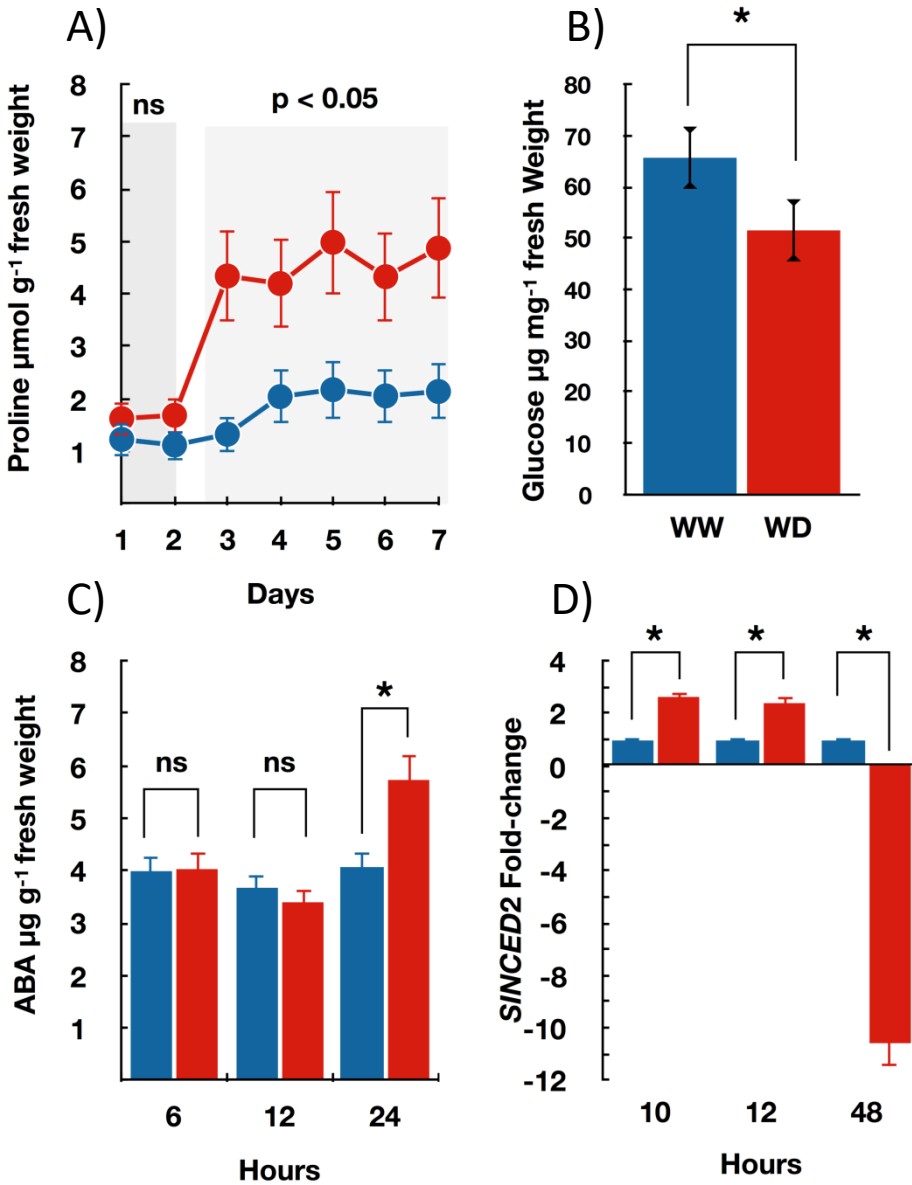

**Figure 6.** Daily (**A**) Proline concentration and (**B**) total carbohydrate concentration in seedlings after 7 growth days. (**C**) ABA accumulation and (**D**) SINCED2 expression. Blue represents well-watered seedlings; red represents water-deficit treated seedlings. * means *p* value < 0.05. ns, not significant.

### 3.3. Method Simplification and Biostimulants Validation

The experimental setup can be transformed into a standard screening methodology by simplifying the data collection. It would be impossible to measure and evaluate all

the variables for every biostimulant molecule or complex mixture formulation. Consequently, the variables used in the experiments to assess water-deficit impact were assessed qualitatively and quantitatively, to achieve that simplification. Table 2 summarises all the conclusions and results. This analysis showed the final dry weight of the seedlings to be the most convenient variable to detect the putative beneficial effect of a specific compound or complex mixture under experimental conditions. The main factors contributing to this decision were: the primary technical abilities, the instrumentation needed to analyse the samples, the time required for this, and the cost per sample during the experiment. In summary, the statistical analysis showed a significant correlation between the seedlings' dry weight and the variables used to describe how water deficit affects gas exchange and total carbohydrates.

**Table 2.** Qualitative and statistical analyses of the variables used to characterise the water-deficit treatments in tomato seedlings under experimental conditions. Dw, dry weight; RGR, relative growth rate; RWC, relative water content; WUEplant, plant water use efficiency; WUE(i:inst).

| Parameters | Variables | Tech./Inst. | Time | Cost * | Correlation ** |
|---|---|---|---|---|---|
| Growth | Dw<br>RGR<br>RWC<br>$WUE_{plant}$ | Low technical abilities from technicians and simple instrumentation | Relative low time, depending on the number of samples and treatments | Low | Dw vs the rest |
| Gas exchange | Pn<br>Gs<br>E<br>WUE(i:inst) | High technical abilities from technicians and expensive instrumentation | Higher time requirements due to the number of samples and treatments | Low | $p < 0.01/0.554$<br>$p < 0.01/0.545$<br>$p < 0.01/0.537$<br>Ns |
| Metabolites/genes | Proline<br>Carbohydrates<br>ABA<br>$SlNC_2$ | High technical abilities from technicians, specific and expensive instrumentation | Demanding, depending on the chosen variable and samples per treatment | High | Ns<br>$p < 0.01/0.548$ |

**Tech./Inst.** Technical abilities and instrumentation. **Cost *** Economic cost per sample analysed. **Correlation ** ** Correlation analysis between Dw and Pn, Gs, E, metabolites.

This approach was validated through analysing seedling dry weight when subjected to water deficit, with known concentrations of the 12 biostimulant compounds (polyamines, amino acids, betaines and inorganic salts) and one seaweed extract (Kelpak®). Figure 7 showed that except for the Kelpak®, all the biostimulants significantly increased the final dry weight of the seedlings. Molecules such as putrescine and spermine (1mM) were the most effective, with a similar RGR to control well-watered seedlings. There was a respective 60% and 45% increase in biomass compared to water-deficit treated seedlings. However, spermidine and glycine-betaine were more efficient (0.1 mM), maintaining slightly lower DW and RGR but using ten times less concentration (0.1 mM). These growth values represent a respective 45% and 35% increment in biomass over the water-deficit treated seedlings. Finally, sometimes in screening processes, access to high amounts of specific compounds or extracts is difficult, which limits testing different concentrations. During the experiments, 5 mL of the tested concentrations of biostimulants were used per plant treated. Nonetheless, this volume can be reduced, maintaining the same effect on seedling growth, expressed as dry weight. Table 3 summarises the results using BABA, a molecule well-known for its effects on plants under biotic and abiotic stresses. The results showed no significant differences between seedlings at the end of the experiment when treated with 1 mL or 5 mL. This result has enormous implications from the economic point of view because this simple modification will significantly reduce the screening costs.

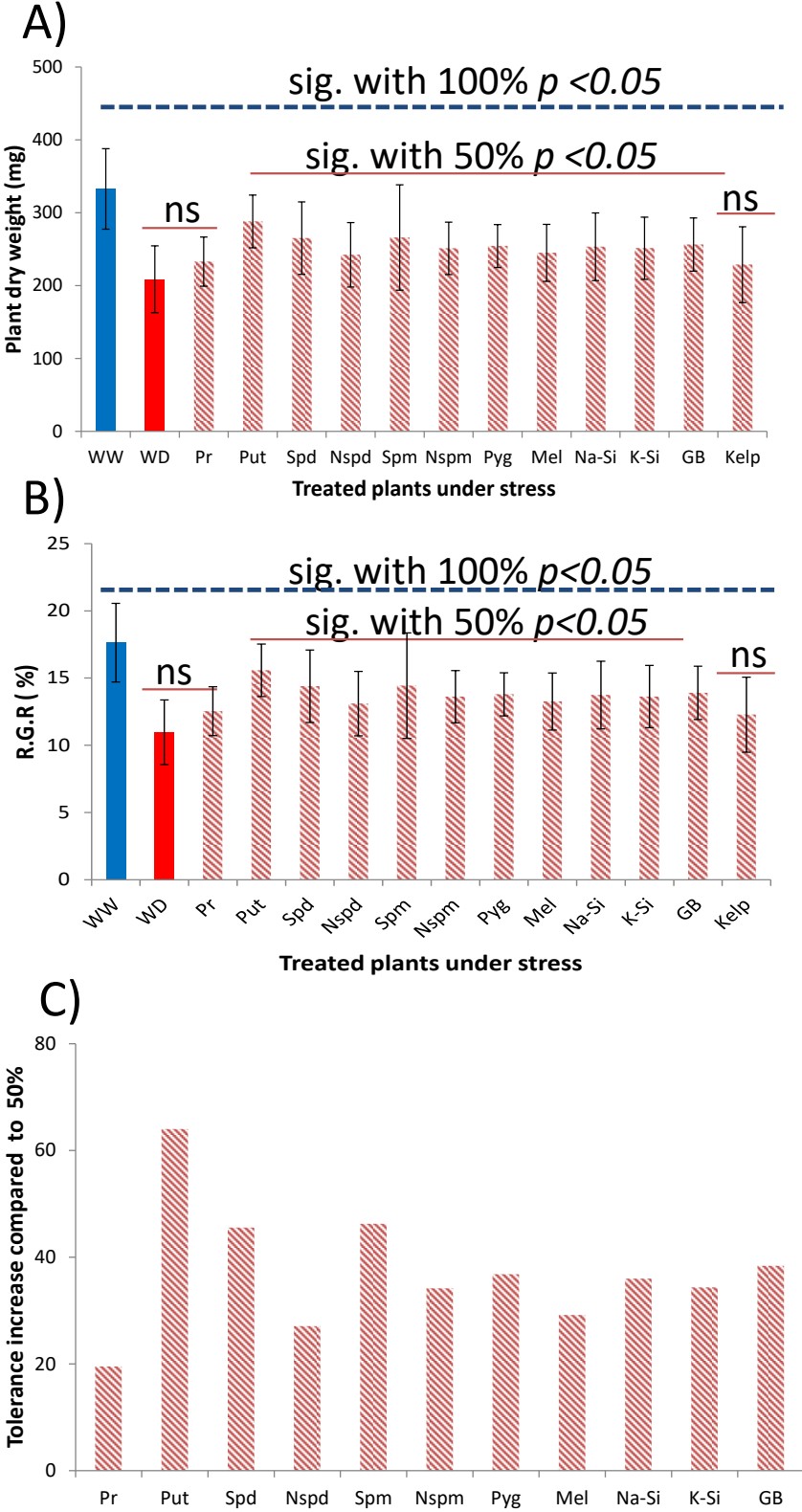

**Figure 7.** (**A**) Biostimulants DW (Dry Weight), (**B**) RGR (Relative Growth Rate) and (**C**) Growth differences (%) compared to water-deficit treated seedlings after seven days of the experiment. Kelpak® was omitted from the third graph because it showed no differences with the water-deficit treated seedlings. ns, not significant.

**Table 3.** Seedlings DW (Dry Weight mg), BABA (mg) and costs (EUR) analysis per treatment depending on the volume (ml) used in the experiments. Different letters represent significant differences between treatments.

|  | Volume (mL) | Mg/Volume | Plants/Treatment * | mg/Treatment * | EUR/Treatment | DW ** |
|---|---|---|---|---|---|---|
| WW | 5 | 0 | - | - | - | 260 ± 25 a |
| WD | 5 | 0 | - | - | - | 193 ± 32 b |
| BABA | 1 | 0.103 | 60 | 6.2 | 0.840 | 255 ± 49 ac |
|  | 3 | 0.309 | 60 | 18.5 | 2.521 | 248 ± 55 ac |
|  | 5 | 0.515 | 60 | 30.9 | 4.202 | 260 ± 21 ac |

BABA (1 mM; 1gr—EUR 136); * Plants used in triplicate experiments; ** Dry mg after 7 days of the experiment.

## 4. Discussion

The screening of biostimulant molecules or formulations for use in a water-deficit scenario requires a new approach. Experimentation should be feasible inside laboratory facilities without special equipment, thus reducing space, time and costs. The challenge was to simplify the methodology to a point where the screening process is facilitated, minimising: (1) the number of plants and the amount of biostimulant used, (2) experimental duration, and (3) the number of variables examined in the process. Standard 150-cell trays were employed, and cell-tray watering capacity was controlled by a reliable and inexpensive method—field water capacity [58].

The first step was to reproduce mild drought effects using a water-deficit scenario in the experimental setup. If the water-deficit treatments last too long, the seedlings will undergo more severe drought stress, minimising the beneficial effects of the biostimulants being assayed. That is the main reason why the experiments were limited to seven days. Once this issue was clarified, the system reproduced similar responses in water-deficit treated seedlings to those encountered in plants under mild drought stress. Previous reports confirm that drought retards cell division and expansion in roots, shoots and leaves, and consequently, seedling and plant growth [59]. Furthermore, stomatal regulation is crucial in supporting photosynthetic capacity in plants under stress conditions. If the plants undergo water-deficit stress, stomatal closure is triggered, and plants prioritise survival over productivity. This quick response, defined as the active closure mechanism, is controlled by a diverse network of signalling pathways, in which the key player is ABA [60]. The first experiment using this method established a significant reduction in biomass (25% average) and gas exchange parameters (Pn, Gs, E and WUEi between 30–50%) in water-deficit treated seedlings, during or at the end of the experimental period. This showed a similar trend to work previously published using this crop [61]. In addition, the imposed water deficit induced *SlNCED2* transcription after only 10 h. This gene encodes 9-cis-epoxycarotenoid dioxygenase, the key enzyme in the ABA biosynthetic pathway [62,63]. This upregulation caused a pronounced ABA accumulation in leaves after 24 h in water-deficit treated seedlings.

Regulation of unfavourable water status is crucial for plant survival under adverse environmental conditions, such as drought. The accumulation of cell-compatible solutes—Osmotic Adjustment (OA)—is generally accepted as the most common reaction to overcome the adverse effects of water deficit [60]. OA increases the concentration of osmotically active solutes in the cell while decreasing the osmotic potential, improving cell hydration and maintaining leaf turgor in metabolically active cells. Therefore, OA is a mechanism for drought tolerance rather than just a drought response [64]. Osmotically active solutes involved in OA include amino acids, ammonium compounds, soluble sugars and plant hormones such as polyamines. All these solutes have numerous hydroxyl groups that assist in facilitating hydrogen bonds with the water molecules in the cytoplasm. Thus, the solutes contribute to osmoregulation and protect the enzymes and macromolecules in cells from the damaging effects of ROS; therefore, operating as antioxidants [65]. Water-deficit treated seedlings reproduced those effects, showing significantly increased proline levels after two days, due to ABA accumulation initiating proline biosynthesis in the water-deficit

treated seedlings [66]. In contrast, total carbohydrates underwent a significant decrease at the end of the experiments, attributable to a low contribution of carbohydrates to OA in young tomato leaves [67].

The second step in developing this methodology involved simplifying the procedure and the number of variables used. Consequently, it significantly increases the number of tested biostimulant molecules or formulations. The variable used to compare and study the list of biostimulants used in this work was the final biomass of the seedlings (dry weight). The selection was made by qualitative and quantitative analysis. This approach was successfully used in a second experiment using the setup described in Figure 1; water-deficit treated seedlings showed, as expected, a significant decrease in biomass compared with well-watered control plants. However, root-treated seedlings with different types and concentrations of biostimulants overcome this effect, reaching similar growth to control plants, depending on the biostimulant used [55]. This significant biomass increase in biostimulant-treated water-deficit seedlings could result from early OA. In fact, previous reports showed how exogenous applications of polyamines or melatonin modulate drought responses in different crops by proline accumulation [68,69]. However, other mechanisms could be involved in biostimulant-treated seedlings under water deficit.

Additionally, this methodology can be adapted to test low amounts of molecules. Using 1 mM of BABA as an example, the methodology demonstrates that 1 mL per plant of this biostimulant induced the same effect on seedling growth as 3 or 5 mL per plant. This change in volumes per treated seedling represents a five-fold difference in mg of BABA used per seedling. This adjustment opens the possibility of increasing the number of new molecules, such as secondary metabolites produced during the normal plant response against biotic and abiotic stresses [61], or new synthetic compounds generally excluded from this kind of evaluation. There are other protocols in the literature to assess biostimulants [44], which focus on transcriptome profiling and field tests. Although our method is perfectly compatible with such techniques, it has the advantage of being cheaper, faster and based on simplicity. It saves resources and time by enabling a fast preliminary screening to detect functional molecules and formulations, which may later undergo more exhaustive field trials or molecular assays, such as transcriptome profiling.

## 5. Conclusions

This report describes a method for screening biostimulants in crop seedlings subjected to a water-deficit scenario. This approach can reproduce mild drought effects in tomato seedlings under water deficit -biomass, total carbohydrates and photosynthesis reduction and an increase in ABA and proline accumulation. A qualitative and quantitative simplification of the variable quantification and data acquisition allowed us to evaluate twelve different biostimulants, with a low budget and technical requirements. Moreover, the method is adjustable to different concentrations and volumes of biostimulants applied to the seedlings. Consequently, it can be used to study and quantify the functionality of different stocks, avoiding methodological errors. Finally, this report provides a new and valuable tool for the biostimulant community, both industrial and academic.

**Supplementary Materials:** The following are available online at https://www.mdpi.com/article/10.3390/agronomy12030728/s1, Table S1: Primers used in the study.

**Author Contributions:** Conceptualization, D.J.-A. and J.C.L.; methodology, D.J.-A. and J.C.L.; investigation, D.J.-A., S.M.-S., A.J.H.; data curation, D.J.-A.; writing—original draft preparation, D.J.-A.; writing—review and editing, J.C.L. and A.A.B.; supervision, A.A.B.; project administration, A.A.B.; funding acquisition, J.C.L. All authors have read and agreed to the published version of the manuscript.

**Funding:** This work was financed by the project AHIDAGRO (MAC2/1.1b/279), Cooperation Programme INTERREG-MAC 2014–2020, with the European Regional Development Fund (FEDER).

**Acknowledgments:** The authors want to dedicate this article to the memory of Francisco Valdes; his passion and joy made this article possible; rest in peace. The manuscript was revised by G. Jones, funded by Cabildo de Tenerife under the TFinnova Programme and supported by MEDI and FDCAN.

**Conflicts of Interest:** The authors declare no conflict of interest.

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
