# Peer review of "New Biostimulants Screening Method for Crop Seedlings under Water Deficit Stress"

_agronomy, doi:10.3390/agronomy12030728_

Round 1

Reviewer 1 Report

The authors have addressed an important substance for water deficiency control. The manuscript in its current form requires many improvements. My comments are as:

  1. The introduction is very short and the latest references from 2021-2022 studies are missing.
  2. The problem statement is not clear in the 1st paragraph.
  3. Why this study was needed? the need for study is missing.
  4. The hypothesis of the study should be added in the introduction section.
  5. How would the biostimulants influence crop production in various climates is missing in the intro section. 
  6. In the methodology section, clearly indicate the source of products used and their manufacturers for the instruments, chemicals, and other inputs. 
  7. References of various techniques in methodology should be added. 
  8. Add a table to clearly differentiate between the two experiments in detail. 
  9.  Why T-test was used, the data should be analyzed with Duncan multiple comparisons test with LSD.
  10. Figures are not conclusive, X-axis has no info indicated. 
  11. Mechanisms are missing in the discussion section especially line 299-302 looks wordy. 
  12.  

Author Response

Dear reviewer 1, many thanks for your useful comments, see the answer below:

  1. The introduction is very short and the latest references from 2021-2022 studies are missing.

The introduction was reorganized adding several passages and references.

  1. The problem statement is not clear in the 1st paragraph.

Crop productivity under global climate change is the problem statement, while biostimulants is part of the answer. The method presented here is a tool for biostimulants screening and formulation.

  1. Why this study was needed? the need for study is missing.

The method is needed for because we have high technological platforms not accessible to all laboratories (in-vitro yeast growth followed by seed germination stage ;Arabidopsis germination and rosette growth using a High-Throughput Screening platform; High-Throughput Plant Phenotyping linked with Metabolomics; using inducible reporters lines, as the ABA-inducible luciferase to search for agonist molecules).  

  1. The hypothesis of the study should be added in the introduction section.

Done

  1. How would the biostimulants influence crop production in various climates is missing in the intro section. 

The intro section is focused in drought stress and water deficiency. The aim was to present a low technical and low cost methodology to test and select biostimulants pure compounds mixtures and formulations. However, two reviews are included in the references:

García-García, A.L.; García-Machado, F.J.; Borges, A.A.; Morales-Sierra, S.; Boto, A.; Jiménez-Arias, D. Pure Organic Active Compounds Against Abiotic Stress: A Biostimulant Overview. Front. Plant Sci. 2020, 11, doi:10.3389/fpls.2020.575829.

Jiménez-Arias, D.; García-Machado, F.J.; Morales-Sierra, S.; García-García, A.L.; Herrera, A.J.; Valdés, F.; Luis, J.C.; Borges, A.A. A Beginner’s Guide to Osmoprotection by Biostimulants. Plants 2021, 10, 363, doi:10.3390/plants10020363.

  1. In the methodology section, clearly indicate the source of products used and their manufacturers for the instruments, chemicals, and other inputs. 

Done

  1. References of various techniques in methodology should be added. 

Done

  1. Add a table to clearly differentiate between the two experiments in detail. 

We modified the Figure 2. Now clearly differentiates both experiments.

  1. Why T-test was used, the data should be analyzed with Duncan multiple comparisons test with LSD.

Done

  1. Figures are not conclusive, X-axis has no info indicated. 

Done

  1. Mechanisms are missing in the discussion section especially line 299-302 looks wordy. 

The discussion section was reorganized, rewritten and possible mechanism for Biostimulants effects on seedlings growth under water deficit stress added.

Reviewer 2 Report

The article requires major changes. My comments are given below.

  • The title is confusing, a simplified and result-oriented title is required.
  • In the abstract section, show the aim of the study.
  • Add some important results in the abstract.
  • It is recommended to add a one-liner conclusion in the abstract. 
  • The aims and objectives should be added at the end of the introduction. 
  • Add 2-3 more paragraphs from recent studies as the introduction loos very small.
  • Beautify the graphs with x-axis color black, bold, and Y-axis mostly with 5-6 values. 
  • The data display is poor needs improvement.
  • 276-288, looks like an assumption, no mechanistic approach is indicated to support the lines. 
  • What is the practical implementation of this study, add a sub-section at the end of the discussion. 
  • Also, add the study limitations, and future research impacts/suggestions. 
  • What is the major conclusion, the current conclusion looks like methodology and discussion.
  • Add the key aspects to this study in conclusion. 
  • The English language needs throughout checking by a native speaker.
  • Check the references with errors than recommended style and missing volume. 

Author Response

Dear reviewer 2, many thanks for your useful comments, see the answer below:

  1. The title is confusing, a simplified and result-oriented title is required.

Done

  1. In the abstract section, show the aim of the study.
  2. Done
  3. Add some important results in the abstract.
  4. Done
  5. It is recommended to add a one-liner conclusion in the abstract.
  6. Done
  7. The aims and objectives should be added at the end of the introduction. 

Done

  1. Add 2-3 more paragraphs from recent studies as the introduction loos very small.

Done

  1. Beautify the graphs with x-axis color black, bold, and Y-axis mostly with 5-6 values. 

We did our best.

  1. The data display is poor needs improvement.
  2. We did our best.
  3. 276-288, looks like an assumption, no mechanistic approach is indicated to support the lines. 

Re-written.

  1. What is the practical implementation of this study, add a sub-section at the end of the discussion. 

Done

  1. Also, add the study limitations, and future research impacts/suggestions. 

This is currently under study in our lab. We can say about it that the same methodology has been used in monocots and dicots crops with similar results. We are using this method in lab to field studies  looking for correlations between biostimulant selection and dose used in water deficit studies. Results are promising.

  1. What is the major conclusion, the current conclusion looks like methodology and discussion.

New conclusion section added.

  1. Add the key aspects to this study in conclusion. 
  2. New conclusion section added.

  1. The English language needs throughout checking by a native speaker.

Done

  1. Check the references with errors than recommended style and missing volume.

Done

Reviewer 3 Report

The literature review is done correctly with the latest citations from the last 20 years. The research methodology is clear to me. I have a question, why were these particular methods used? The results are presented clearly. Maybe it's worth using a post-hoc test and name it? I don't mind Excel charts but they look a bit schooly. The word discussion - capital letter. please correct the text editing. Discussion to analyze and reformat the conclusions. Working on the language - native speaker. 

Author Response

Dear reviewer 3, many thanks for your useful comments, see the answer below:

  1. I have a question, why were these particular methods used? 

The method is needed for because we have high technological platforms not accessible to all laboratories (in-vitro yeast growth followed by seed germination stage ;Arabidopsis germination and rosette growth using a High-Throughput Screening platform; High-Throughput Plant Phenotyping linked with Metabolomics; using inducible reporters lines, as the ABA-inducible luciferase to search for agonist molecules). Although this methodology is perfectly compatible with such methods, our method has the advantage of being cheaper, faster and based on simplicity

  1. Maybe it's worth using a post-hoc test and name it? 

Done

  1. I don't mind Excel charts but they look a bit schooly. 

We did our best.

  1. The word discussion - capital letter. please correct the text editing. 

Done

  1. Discussion to analyze and reformat the conclusions. 

Done

  1. Working on the language - native speaker.

Done

Round 2

Reviewer 1 Report

Article is improved, but can be accepted after data presentation improved, figures are good enough for a journal like agronomy. 

Author Response

Many thanks for your comments, we had a problem with figures 1 and 2 in the previous version, but now are solved. We improved the others too. We hope you like the new ones.

Reviewer 2 Report

The authors have improved the article significantly, still Figures are not very well beautified to attract the reader. The figures needs to be modified, make them elegant. 

Author Response

Thanks for the comments and we agree with you, we had a problem with the previous version on figures 1 and 2, now are fixed. We improved the other too, we hope you like the new ones